# Embracing Adaptation: An Effective Dynamic Defense Strategy Against Adversarial Examples

Shenglin Yin
Peking University
School of Computer Science
Beijing, China
yinsl@stu.pku.edu.cn

Kelu Yao
Zhejiang Lab
Zhejiang University
Hangzhou, China
yaokelu@zhejianglab.com

Zhen Xiao*
Peking University
School of Computer Science
Beijing, China
xiaozhen@pku.edu.cn

Jieyi Long
Theta Labs, Inc
San Jose, USA
jieyi@thetalabs.org

## ABSTRACT

Existing adversarial example defense methods are static, meaning they remain unchanged once training is completed, regardless of how attack methods change. Consequently, static defense methods are highly vulnerable to adaptive attacks. We argue that to counter more formidable attacks, models should continually adapt to various attack methods. We propose a novel dynamic defense approach. Initially, we use Gaussian Mixture Models (GMM) to obtain structural information of the data, which is combined with model prediction information to generate pseudo-labels for optimizing inputs. Subsequently, we employ information maximization and enhanced mean predictions as optimization objectives, utilizing a hierarchical optimization approach to refine the model. Meanwhile, we propose a sample-efficient optimization strategy that reduces the total number of samples in the test data stream for reverse updating and improves the efficiency. Notably, our method can be directly applied to pre-trained models without the need for accessing training data or retraining the model. Therefore, our approach is training-data-agnostic and model-agnostic, easily applicable to existing adversarially trained models, significantly enhancing the resilience of various models against white-box, black-box, and adaptive attacks across diverse datasets. We have conducted extensive experiments to validate the state-of-the-art of our proposed method. *The pseudo-code can be found in the appendix.*

## CCS CONCEPTS

• **Computing methodologies** → **Computer vision**.

---

*Corresponding author.

## KEYWORDS

adversarial attack, dynamic defense, robustness

**ACM Reference Format:**
Shenglin Yin, Kelu Yao, Zhen Xiao, and Jieyi Long. 2024. Embracing Adaptation: An Effective Dynamic Defense Strategy Against Adversarial Examples. In *Proceedings of the 32nd ACM International Conference on Multimedia (MM '24), October 28-November 1, 2024, Melbourne, VIC, Australia.* ACM, New York, NY, USA, 10 pages. https://doi.org/10.1145/3664647.3680580

## 1 INTRODUCTION

Deep Neural Networks (DNNs) have achieved significant success in both academic and industrial applications, including image classification [18], face recognition [48], time series forecast [52] and resource scheduling [53, 54]. However, DNNs are vulnerable to the threat of adversarial examples. The attacker and the defender are two players in game theory, and when they are both deterministic, there is no Nash Equilibrium between them [37]. This gives rise to a recurring phenomenon in contemporary deep learning-based adversarial defenses. Adversarial learning research engages in a cat-and-mouse game between the attacker and defender, where a new attack is proposed, followed by a subsequent defense to mitigate it. This cycle continues as the attacker devises new attacks that exploit vulnerabilities in the previous defense, perpetuating an ongoing pattern. To effectively counter more potent adaptive attacks, defenders must dynamically adapt their defenses in response to the attacker's evolving strategies, thereby enabling victory in the game. However, the majority of existing defense methods remain static [4, 9, 14, 30, 32], meaning that they do not alter the model's parameters post-training and are unable to adapt to evolving attacker tactics (see Figure 1 (a)).

In recent years, researchers have been investigating dynamic defense techniques. However, many of these approaches have inherent limitations. Some research [8, 25] efforts focus on adapting model structures so that they can autonomously modify their network states during the inference phase. Additionally, there are approaches [34, 41] that employ extra purification modules during inference to transform inputs and lessen the effects of adversarial disturbances. However, these methods may not always be practical. A primary issue is their reliance on original training data, which poses challenges due to privacy concerns. For example, in

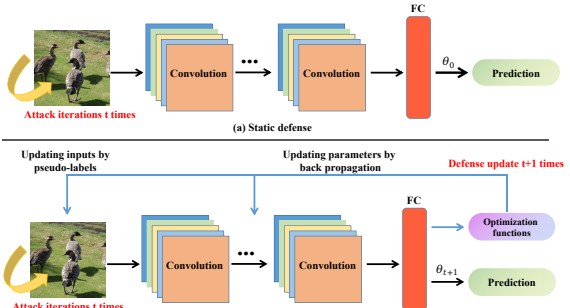

**Figure 1: In the static defense (a), the attacker optimizes the input against the model parameter $\theta_0$ and the model uses parameter $\theta_0$ to make predictions about input $\mathbf{x}_t$. In our proposed dynamic defense (b), the input and parameters are optimized during inference, and the model uses parameter $\theta_{t+1}$ to make predictions for input $T_{t+1}(\mathbf{x}_t)$.**

sectors like healthcare, federated learning is used to maintain data security during model training. In such scenarios, accessing the complete training dataset is not feasible, and techniques like adversarial training may not ensure model robustness. Furthermore, retraining models or training new purification modules with large datasets can be resource-intensive. The Dent [46] optimizes models and inputs by reducing input entropy, while the Anti [1] introduces disturbances to inputs to thwart attackers. Notably, neither method requires the original training set. These self-training methods have been effective with data from static domains. However, their stability is compromised when facing adversarial examples from continuously evolving attackers [38, 49]. This instability is due to increased pseudo-label noise and miscalibration caused by attacks [16], leading to shifts in the attack distribution. Consequently, early predictive errors are more likely to accumulate [6], resulting in defense failures. Moreover, these solutions address issues in a single dimension, which can harm the stability of the methods and their effectiveness against adaptive attacks.

To address these issues, this paper proposes a dynamic defense method that is training-data-agnostic and model-agnostic, addressing the limitations of existing technologies through input and model optimization. As shown in Figure 1(b), the goal is to start from a ready-made pre trained model and always protect the model from the impact of constantly changing adversarial examples. For input optimization, we use Gaussian Mixture Model (GMM) to extract input structural information at the feature layer and combine this information with model prediction information to generate pseudo labels. Subsequently, projection gradient descent is used to move the input as much as possible towards the correct region. As for model optimization, considering significant distribution differences between adversarial examples and raw data, a weighted average teacher model is adopted to enhance prediction accuracy. Specifically, we perform multiple stochastic augmentations of the inputs and predict the augmented inputs using the teacher model, and then average the multiple predictions. At the same time, in order to reduce the accumulation of errors caused by model prediction errors, adaptive sample weights were introduced to guide the training of student models, and a hierarchical optimization method was used

to stabilize the entire optimization process. Furthermore, it should be noted that input optimization and model optimization may involve excessive forward and backward propagation, resulting in additional time consumption. To alleviate this situation, we propose a sample efficient optimization strategy that excludes highly confident samples and redundant samples from the model during optimization. In this case, the total number of reverse updates in the test data stream is appropriately reduced (improving efficiency), and the performance of the model on adversarial examples is also improved. The contributions of our work are as follows:

- We propose a dynamic defense strategy that adjusts inputs themselves and model parameters during the inference phase in response to the current inputs of the model.
- We propose a method that is agnostic to training data and model architecture, requiring no modifications to the model structure or retraining. It can be directly applied to pre-trained models.
- Extensive experiments have demonstrated that our method outperforms state-of-the-art methods in terms of improving the model's accuracy on adversarial examples.

## 2 RELATED WORK

### 2.1 Adversarial Attacks

Sezgedy et al. [43] first introduced the concept of adversarial examples, i.e., adding noise that is imperceptible to the human eye to the original clean examples, so that the perturbed examples cause DNNs prediction errors. After this concept was introduced, many studies investigated the robustness of the model and proposed a series of attack methods. Goodfellow et al. [14] proposed a method to generate adversarial examples using Fast Gradient Sign Method (FGSM), which finds the most aggressive perturbation within a fixed range of perturbations by exploiting the gradient information of the model. Madry et al. [32] further improved FGSM by proposing a multi-step version of FGSM called projected gradient descent (PGD). It generates the adversarial examples by randomly perturbing the original samples in their neighborhood as the initial input, and then generating adversarial examples after several iterations. Carlini and Wagner [5] proposed the C&W attack method for the defense distillation network proposed by Hinton et al. [20]. The C&W method is divided into three categories according to the attack target category: random targets, the easiest category to attack, and the hardest category to attack, and the perturbation is optimized by restricting the $l_0, l_2, l_\infty$ parametrization. Croce et al. [13] first proposes two extensions of the PGD attack to overcome failures due to problems with suboptimal step sizes and objective functions, and then combines the new attacks with two complementary existing attacks to form a parameter-free, computationally tolerable and user-independent combination of attacks to test the robustness of the adversarial. This approach is called AutoAttack.

### 2.2 Adversarial Training

Adversarial training is one of the most effective ways to counteract adversarial examples [33]. Athalye et al. [3] show that most defense methods are ineffective against gradient mask-based adaptive adversarial attacks, and that adversarial training is the only defense

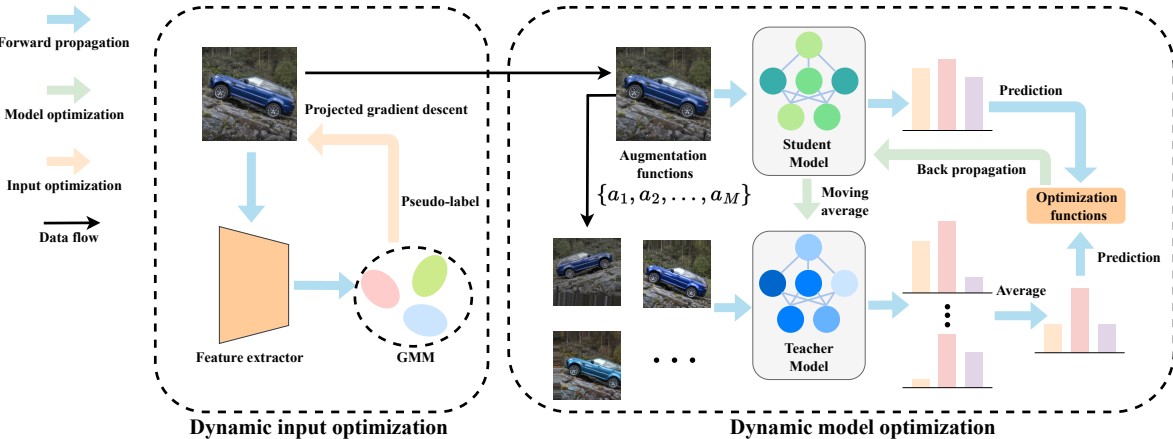

**Figure 2: The overall framework of the proposed method. We first optimize the input by generating pseudo-labels via the GMM algorithm. The teacher model guides the student model to adjust its own parameters based on the optimized input.**

that has been shown to be effective. Goodfellow et al. [14] first introduced the concept of adversarial training, which trains a model by adding adversarial examples to the training set. That is, the model needs to deal with both clean samples and auxiliary adversarial examples. Madry et al. [32] proposed a framework for primary adversarial training (PGD-AT) to improve the robustness of the model. As representative work on adversarial training, PGD-AT has had a far-reaching impact. However, it significantly reduces the accuracy of clean samples and has a computational cost that far exceeds the cost of standard training. Zhang et al. [56] proposed TRADES that trade-offs adversarial robustness and accuracy by decomposing the prediction error (robustness error) of the adversarial example into natural (classification) error and boundary error. Wu et al. [51] design Adversarial Weight Perturbation (AWP) to enhance the robust generalization of the model. AWP adds adversarial perturbation not only to the input samples but also to the parameters of the model, and demonstrates that adversarial weight perturbation does lead to a tight upper bound on robust generalization. Jia et al. [23] introduced the concept of learnable attack strategies, proposing a new adversarial training framework that learns to automatically generate attack strategies to improve the robustness of the model.

### 2.3 Dynamic Defense

Shi et al. [41] additionally design a self-supervised task during the training phase of the model to clean up adversarial perturbations by minimizing auxiliary losses during inference. Mao et al. [34] point out that images contain intrinsic structure that can reverse adversarial attacks. They used a self-supervised task trained in advance to purify the input, and the representation exploited the intermediate activation of a pre-trained static model. Yoon Jongmin et al. [55] preprocess the input to the classifier, train it with an energy-based model (EBM) and use denoised score matching (DSM) to learn a score function to denoise the scrambled images. Hwang et al. [21] propose the use of a discriminator to cleanse the input of a pre-trained classifier. The discriminator is trained to distinguish between adversarial perturbations and cleaned inputs. However, these methods require access to the original training dataset, which is not applicable in many cases. When the amount of data is large,

additional training aids or retraining the entire model is very time consuming as well as wasteful of resources. Dent [46] links adversarial defense to domain adaptation, which uses defense updates to counter attack updates. As the adversary optimizes across the decision boundary, entropy minimization optimizes the distance between the prediction and the decision boundary, thus disrupting the attack. Anti [1] uses the model's predictions as pseudo-labels and adds perturbations to inputs using projected gradient descent to move inputs away from the decision boundary, thus defending against attackers. Although both methods do not rely on training data, the pseudo-labels they generate lack credibility and can result in error accumulation, ultimately compromising the effectiveness of the defense. Most of these methods were defeated by the adaptive attack devised by Croce Francesco et al. [11].

## 3 METHOD

### 3.1 Preliminary

Let $\mathbf{x} \in \mathbb{R}^d$ be the input image and $y \in \mathbb{R}^C$ be the label. Given the model $f_\theta(\cdot) : \mathbb{R}^d \to \mathbb{R}^C$ parameterized by $\theta$ and a clean image $\mathbf{x}$, the goal of attacker is to create an adversarial image $\mathbf{x}_{adv} = \mathbf{x} + \boldsymbol{\delta}$ that is similar to clean image but confuses $f_\theta(\cdot)$:

$$
\begin{aligned}
d(\mathbf{x}_{adv}, \mathbf{x}) &< \epsilon, \\
f_\theta(\mathbf{x}_{adv}) &\neq f_\theta(\mathbf{x}),
\end{aligned}
\tag{1}
$$

where $d(\cdot, \cdot)$ is the distance function between the clean image and the adversarial image. $\epsilon$ is the perturbation scale and is usually set to a small number to obtain an almost imperceptible difference between $\mathbf{x}_{adv}$ and $\mathbf{x}$.

In dynamic defense, we have access to a pre-trained model $f_\theta(\cdot)$ with parameters $\theta$ trained by adversarial training on a clean dataset. Without loss of generality, our goal is to improve the robustness of the model against various types of attacks in an online fashion during the inference. At time step $t$, the unlabeled test data $\mathbf{x}_t$ is provided as input. We first adjust $\mathbf{x}_t$ and then the model $f_{\theta_t}(\cdot)$ needs to adjust itself according to the adjusted $T_{t+1}(\mathbf{x}_t)$, where $T(\cdot)$ is the method of adjusting the input, and finally adjusted model $f_{\theta_{t+1}}(\cdot)$

needs to make prediction $f_{\theta_{t+1}}(T_{t+1}(\mathbf{x}_t))$. The overall process of the proposed method is shown in Figure 2.

## 3.2 Dynamic Input Optimization (DIO)

The goal of an attacker is to maximize the loss function of the input $\mathbf{x}$ by moving it closer to the decision boundary, which results in a prediction with less confidence in the correct label. Thus, it is crucial to move as many samples as possible back to the correct region before making predictions on them. However, this is difficult to do accurately without labels.

Motasem Alfarra et al. [1] proposed a method that uses the model's predictions as pseudo-labels and applies projected gradient descent to generate a new input $(\mathbf{x} + \hat{\boldsymbol{\delta}})$ that moves $\mathbf{x}$ away from the decision boundary, which can defend against attackers. However, relying solely on model predictions as pseudo-labels is unreliable, particularly when it comes to adversarial examples. Additionally, accumulating prediction errors can affect the overall effectiveness of the model [24].

We draw inspiration from pseudo-label generation methods in unsupervised domain adaptation and employ spatial feature clustering to generate more reliable pseudo-labels. Specifically, we utilize GMM to cluster the input features and assign pseudo-labels to the inputs. Compared to other clustering methods, GMM outperforms other clustering methods in terms of confidence because it offers soft labels for the inputs, which are probabilities assigned to the data structure [28, 29]. Therefore, we add the data structure information obtained through the GMM to the model prediction results to improve the reliability of the pseudo-label.

First, we divide the model $f_{\theta_0}(\cdot)$ at time $t = 0$ into a feature extractor $h(\cdot) : \mathbb{R}^d \to \mathbb{R}^k$ and a classifier $g(\cdot) : \mathbb{R}^k \to \mathbb{R}^C$. The feature extractor $h(\cdot)$ is used to obtain the features $\mathbf{A}_t \in \mathbb{R}^k$ for GMM clustering. Perform one EM iteration of the GMM to obtain the probability $p(\mathbf{A}_t) = \{p(\mathbf{A}_t)_1, p(\mathbf{A}_t)_2, ..., p(\mathbf{A}_t)_C\}$ of $\mathbf{A}_t$ over each class, where $p(\mathbf{A}_t)_c = p(y = c|\mathbf{A}_t)$. Finally, we obtain the pseudo-label $y_p$ by the following equation:

$$y_p = argmax(f_{\theta_0}(\mathbf{x}_t) + \gamma p(\mathbf{A}_t)), \tag{2}$$

where $\gamma$ is the hyperparameter used to weigh the structural information. After obtaining pseudo-labels, we follow the PGD approach [32] and adjust inputs.

## 3.3 Dynamic Model Optimization (DMO)

Optimizing a model during inference without input labels presents a challenging task. We have observed a notable distribution difference between adversarial examples and original samples (see Figure 3(b)). As a result, we approach the problem from a different perspective: *How can we reduce the discrepancy between adversarial and original examples?* One solution, proposed by Dent, involves reducing the model's entropy. However, relying solely on entropy reduction for model optimization may not be entirely reliable.

We draw inspiration from the success of test-time augmentation [42] in improving model robustness and the superior accuracy of weight-averaged models over final models [44, 49]. To improve our model's robustness, we augment inputs and leverage the weight-averaged teacher model to generate pseudo-labels, which we use to guide the student model during tuning (see Figure 2). At time

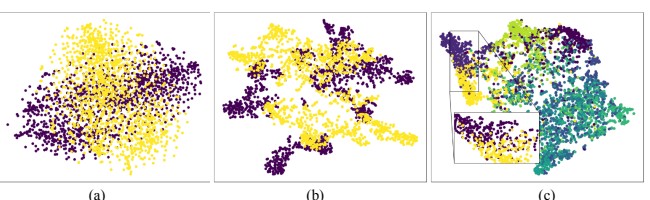

**Figure 3: Visualizing feature using t-SNE. In (a) and (b), purple dots represent adversarial examples and yellow dots represent normal original. In (c), different colored dots represent different categories of samples. (a) Clustering results for the adversarial examples and original samples at the first layer of the model's feature output; (b) Clustering results for the adversarial examples and original samples at the fourth layer of the model's feature output; (c) Feature clustering results for adversarial examples under an adversarial training model.**

step $t = 0$, we use the original model $f_{\theta_0}(\cdot)$ to initialize the teacher model $f_{\theta_0}^T(\cdot)$ and the student model $f_{\theta_0}^S(\cdot)$ .

Furthermore, if we can mitigate the distribution differences between adversarial examples and original samples, the classification output of an unlabeled sample should be similar to that of the original sample, but each class should still be distinct [31]. We achieve this by employing information maximization (IM) loss [26, 31, 45], which ensures that the target output is both individually deterministic and globally diverse.

In practice, we minimize the following two loss functions, $L_{S-T}$ and $L_{IM}$, which together comprise the final loss for tuning the student model:

$$L_{S-T}(\mathbf{p}, \mathbf{p}') = -\sum_{c=1}^{C} S_c(\mathbf{p}) \log S_c(\mathbf{p}'), \tag{3}$$

$$L_{IM}(\mathbf{p}) = -\sum_{c=1}^{C} S_c(\mathbf{p}) \log S_c(\mathbf{p}) + \sum_{c=1}^{C} \hat{p}_c \log \hat{p}_c, \tag{4}$$

where $\mathbf{p} = f_\theta(\mathbf{x})$. $S_c(\mathbf{a}) = \frac{exp(a_k)}{\sum_i exp(a_i)}$ denotes the c-th element in softmax output of a C-dimensional $\mathbf{a}$, $\hat{p} = \mathbb{E}[S(\mathbf{p})]$ is the mean output embedding of the input domain.

While DIO can provide high-quality pseudo-labels, it still has some limitations. According to the clustering assumption [15], data in the same cluster should be assigned the same label. However, the decision boundary of the model may not conform to the clustering assumption [28]. As shown in Figure 3(c), some clusters may contain samples with different ground truth labels. Assigning the same pseudo-label to all samples in a cluster based on the clustering assumption can result in incorrect pseudo-labels, which can negatively impact model optimization results. To learn robustly under samples with incorrect pseudo-labels, we need to suppress samples with low confidence in their pseudo-labels during model optimization.

To solve this problem, we assign a weight to each input. We calculate the $DIFF(\mathbf{x}_t) = p(H(\mathbf{x}_t))_{max} - p(H(\mathbf{x}_t))_{mean}$ for each sample, where $p(H(\mathbf{x}_t))$ is obtained from the GMM. Here, $p(H(\mathbf{x}_t))_{max}$ is the largest value in $p(H(\mathbf{x}_t))$, and $p(H(\mathbf{x}_t))_{mean}$ is the mean value

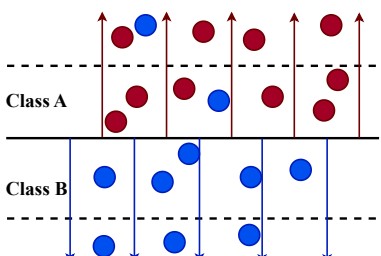

**Figure 4: When the attacker makes the sample cross the correct classification boundary, the input optimization causes errors to accumulate, moving it further and further away from the correct classification boundary.**

of the sum of the probabilities of the other regions. We calculate the weight $W$ for each sample using the following equation:

$$W(\mathbf{x}_t) = \begin{cases} 1 & \text{if } \frac{DIFF(\mathbf{x}_t)}{DIFF_{\max}} \geq \alpha \\ \max\left(\frac{DIFF(\mathbf{x}_t)}{DIFF_{\max}}, \beta\right) & \text{else} \end{cases}, \quad (5)$$

where $DIFF_{\max}$ is the maximum value in each batch and $\alpha$ is the threshold value. For the input far from the GMM-based decision boundary, $W$ will have a high value. Conversely, for the input at the decision boundary, $W$ will have a small value. We weight the loss calculated by $W$ for the input using Eq. 3 and Eq. 4, and thus obtain the final optimization objective.

$$L_{S-T}(\mathbf{p}, \mathbf{p}'; W) = -W \sum_{c=1}^{C} S_c(\mathbf{p}) \log S_c(\mathbf{p}'), \quad (6)$$

$$L_{IM}(\mathbf{p}; W) = -W \sum_{c=1}^{C} S_c(\mathbf{p}) \log S_c(\mathbf{p}) + \sum_{c=1}^{C} \hat{p}_c \log \hat{p}_c, \quad (7)$$

$$L = L_{S-T}(\hat{\mathbf{p}}^S, \hat{\mathbf{p}}^T; W) + L_{IM}(\hat{\mathbf{p}}^S; W) + L_{IM}(\hat{\mathbf{p}}^T; W), \quad (8)$$

where $\hat{\mathbf{p}}^S = f_{\theta_t}^S(\mathbf{x}_t)$ and $\hat{\mathbf{p}}^T = \frac{1}{N_a} \sum_{i=1}^{N_a} f_{\theta_t}^T(aug_i(\mathbf{x}_t))$ are the predicted labels of the student model and the teacher model at time $t$, respectively. After updating the student model from $\theta_t^S$ to $\theta_{t+1}^S$ using Eq. 8, we update the weights of the teacher model by taking a weighted average of the student model weights. We use an Exponential Moving Average (EMA) to compute the weighted average and use a smoothing factor $\lambda$ to control the trade-off between the teacher and student models. Finally, we use the teacher model for the final prediction.

$$\theta_{t+1}^T = (\lambda - 1)\theta_t^T + \lambda\theta_{t+1}^S. \quad (9)$$

### 3.4 Sample Filtering (SF)

To effectively adapt to test time and avoid the accumulation of errors in the model optimization process, we introduced an active sample identification strategy for selecting samples during the backpropagation process. This strategy involves assigning a selection score $I(\mathbf{x})$ to each sample. By setting $I(\mathbf{x}) = 0$, samples that do not participate in backpropagation are eliminated, thereby reducing unnecessary backward calculations during test time adaptation and enhancing prediction efficiency.

**Overconfident samples.** In the face of adversarial attacks, even with the optimization of inputs using high-quality pseudo labels,

some samples may still cross classification boundaries. In an unlabeled test environment, input optimization guided by incorrect pseudo labels can lead to error accumulation, causing samples to deviate from the correct classification boundary. This deviation makes the model overly confident in its predictions, which may actually be incorrect (as shown in Figure 4). To prevent the model from optimizing using such samples, we propose an entropy-based scheme for identifying reliable samples. Formally, the entropy-based scheme is given by:

$$I_{os}(\mathbf{x}) = \mathbb{I}_{\{E(\mathbf{x};\theta) > E_0\}}(\mathbf{x}), \quad (10)$$

where $I_{\{\cdot\}}(\cdot)$ is the indicator function, $E(\mathbf{x};\theta)$ is the entropy of sample $\mathbf{x}$, and $E_0$ is a predefined threshold. Entropy indicates the model's uncertainty; a higher entropy value means greater uncertainty about the prediction, and vice versa. The above function optimizes the model by excluding overly confident samples. It is noteworthy that evaluating $I(\mathbf{x})$ does not require gradient backpropagation.

**Redundant samples.** Eq. 10 eliminates some unreliable samples, but there may be redundancy in the remaining test samples. For example, if the predictive entropy of two similar test samples is lower than $E_0$, it is still necessary to perform backpropagation with Eq. 9 for each sample. However, this may be redundant because similar samples produce similar gradients [35]. To improve efficiency, we recommend using samples that produce different gradients for model adaptation. Recall that since the true labels are not available in the inference phase, Eq. 9 relies only on the final output of the model (i.e., classification logits). We further filter the samples by ensuring that the model outputs of the remaining samples are dissimilar. Specifically, for one of the $N$ outputs, we find the most similar from the remaining $N-1$ elements, generate $\frac{N \times (N-1)}{2}$ similarity pairs, and remove the elements that occur more than $K$ times in these pairs. When the number of samples in a batch is large, the time complexity of computing the similarity between features is high and usually unacceptable in practical applications. To reduce the complexity, we use the Ball-Tree algorithm to compute the nearest neighbors of each element, which can be easily achieved by sklearn's NearestNeighbors[1].

$$\{\mathbf{m}_i\}_{i=1}^{N} = NearestNeighbors(f(\{\mathbf{x}_i\}_{i=1}^{N})),$$
$$I_{rs}(\mathbf{x}) = \mathbb{I}_{\{Count(\{\mathbf{m}_i\}_{i=1}^{N}) < K\}}(\mathbf{x}), \quad (11)$$

where $\{\mathbf{x}_i\}_{i=1}^{N}$ is the input batch of $N$ samples, and $\{\mathbf{m}_i\}_{i=1}^{N}$ is each sample corresponding to the sample with which it is most similar. $Count(\cdot)$ is a count of the number of times each sample in m appears.

The overall sample-filtering is then given by:

$$I(\mathbf{x}) = I_{os}(\mathbf{x}) \cdot I_{rs}(\mathbf{x}), \quad (12)$$

which incorporates both the entropy-based filtering Eq. 10 and the diversity-based filtering Eq. 11 The efficiency of the algorithm is further improved since we only perform gradient backpropagation on test samples with $I(\mathbf{x}) = 1$.

---

[1]https://scikit-learn.org/stable/modules/generated/sklearn.neighbors.NearestNeighbors.html#sklearn.neighbors.NearestNeighbors

**Remark.** Given a batch containing $N$ test samples $D_{test} = \{\mathbf{x}_i\}_{i=1}^N$, the total number of reduced inverse computations is given by

$$\mathbb{E}_{\mathbf{x} \sim D_{test}} \left[ \mathbb{I}_{\{I(\mathbf{x})=0\}}(\mathbf{x}) \right] \tag{13}$$

which is jointly determined by the distribution of test data $D_{test}$, the entropy threshold $E_0$, and the number of repetitions $K$.

### 3.5 Optimization Process

During the process of optimizing the model parameters, typically the parameters of all layers are adjusted simultaneously. However, the effect of adversarial perturbations on each layer of the model is different, becoming more severe with increasing depth of the layers (see Figure 3(a) and (b)). If adversarial perturbations can be mitigated in the earlier layers, then the features can be better recovered for the later layers. Therefore, adjusting all layers at the same time can potentially harm the learning of features in the later layers. To address this issue, we propose a hierarchical layer-wise training strategy. Specifically, we first train only the parameters of the first layer for several epochs, and then gradually allow the parameters of the subsequent layers to be trained until the parameters of the final layer are also trained. This approach helps to ensure that the later layers can learn robust features based on the earlier layers' learned robust features.

## 4 EXPERIMENTS

We evaluate proposed method against white-box, black-box, and adaptation attacks using various adversarial training methods and datasets. For attacks, we used PGD (PGD-20 and PGD-50) [32], C&W [5], RayS [7], adaptive attacks designed for the proposed method, and AutoAttack [13]. AutoAttack includes four types of attacks, including white-box attacks (APGD and FAB [12]) and black-box attack (Square attack [2]). For datasets, we use CIFAR10/CIFAR-100 [27], which are often used in studies of adversarial robustness. For defense methods, on the CIFAR-10 dataset, we use the methods proposed by Sehwag Vikash et al. (ResNet-18) [40] and Jingfeng Zhang et al. (WideResNet-28-10) [57]; for CIFAR100, we use the methods proposed by Rice Leslie et al. (PreActResNet-18) [39] and Hendrycks Dan et al. (WideResNet-28-10) [19]. All pre-trained models are obtained from RobustBench [10].

**More exploratory experimental results can be found in the Appendix.**

### 4.1 Competitive Methods

To evaluate the effectiveness of our proposed method, we conducted a comparison with the current mainstream benchmark methods, namely Dent [46] and Anti [1]. We specifically chose these two methods because they share a consistent setup with ours: they do not require access to a training dataset or modification of the model's training method, and can be directly applied to the pre-trained model. It is important to note that most other dynamic defense methods do not meet these criteria. In instances where no defense method was used, we denote it as "None".

### 4.2 Implementation Details

Throughout our experiments, we set the hyperparameters of our method to the following values: number of sample optimization iterations $N_{DIO} = 2$, step size $\zeta = 0.15$, trade-off parameter $\gamma = 0.05$, number of model optimization iterations $N_{DMO} = 2$, trade-off parameter $\lambda = 0.01$, threshold values $\alpha = 0.1$, $\beta = 0.8$, entropy threshold $E_0 = 0.5$ and the number of repetitions $K = 2$. We use SGD as an optimizer with a learning rate of 1e-3. For comparison methods used in our experiments, we followed the settings outlined in their respective papers. We established a maximum perturbation strength of 8/255 for all attack methods under $L_\infty$. For PGD, we implemented a step size of 2/255. For C&W, we utilized the acceleration optimization suggested by Zhang et al. [56]. Finally, for RayS, we randomly selected 3000 samples from CIFAR10/100 and executed 1000 queries per sample.

### 4.3 Evaluation

We present the accuracy results of various defense methods on both clean and different adversarial examples. Additionally, we utilize weighted robust accuracy [17] as an evaluation metric to measure the trade-off between the accuracy of clean samples and that of different adversarial examples. This metric is defined as follows:

$$AVG = \gamma_0 Acc_{clean} + \gamma_1 \left( Acc_{adv_0} + ... + Acc_{adv_N} \right), \tag{14}$$

where $\gamma_0 = \gamma_1 = \frac{1}{N+2}$. $Acc_{clean}$ represents the accuracy on clean samples and $(Acc_{adv_0}, ..., Acc_{adv_N})$ represent the accuracy under different attacks, respectively. It means that both the accuracy of clean samples and the accuracy of different adversarial examples as equally important for assessing the overall performance of the model. This approach emphasizes the importance of developing defense methods that are effective against multiple types of attacks, rather than solely focusing on one or a few attacks.

### 4.4 Evaluation on White-box and Black-box Attacks

We conducted experiments on the CIFAR-10 and CIFAR-100 datasets using different methods and models. Table 1 presents the results of our experiments, indicating that our method effectively enhances the model's robustness against various attacks on both datasets. For instance, our method resulted in a 24.27% improvement in robustness under PGD-50 attack for the ResNet-18 model on the CIFAR-10 dataset compared to the undefended model. It also yielded a 6.78% improvement over the Dent method. On the CIFAR-100 dataset, our method displayed a significant enhancement in robustness as well. Our method achieved the best results in terms of average robust accuracy, demonstrating its ability to maintain strong performance even when faced with different types of attacks. More importantly, Anti decreased the robustness of the model when exposed to the RayS attack. Additionally, Dent's performance on small model (ResNet-18) also suffered, resulting in a reduction of the overall robustness of the model. In contrast, our method consistently enhances the robustness of the model on both large and small models, without any adverse effects observed.

**Table 1: White-box and black-box attacks on the CIFAR-10/100. The best results are boldfaced, and the second best results are** underlined.

| Dataset | Method | Defense | Clean | PGD-20 | PGD-50 | C&W | RayS | AVG |
|---|---|---|---|---|---|---|---|---|
| CIFAR-10 | ResNet-18 [40] | None | **84.59%** | 58.99% | 58.83% | 57.39% | 70.73% | 66.11% |
| | | Dent | 84.25% | 76.01% | 76.32% | 72.78% | 70.70% | 76.01% |
| | | Anti | 84.55% | 81.00% | 80.89% | 81.18% | 70.67% | 79.66% |
| | | Proposed | 84.57% | **83.10%** | **83.03%** | **81.30%** | **72.90%** | **80.98%** |
| | WideResNet-28-10 [57] | None | **89.36%** | 67.77% | 67.67% | 60.57% | 76.43% | 72.36% |
| | | Dent | 89.16% | 80.65% | 80.56% | 78.53% | 78.00% | 81.38% |
| | | Anti | 89.34% | 84.42% | 84.34% | 84.60% | 76.20% | 83.78% |
| | | Proposed | 89.35% | **86.51%** | **86.40%** | **85.58%** | **78.30%** | **85.23%** |
| CIFAR-100 | PreActResNet-18 [39] | None | **53.83%** | 21.16% | 20.94% | 20.59% | 31.23% | 33.74% |
| | | Dent | 53.59% | 40.52% | 40.94% | 39.62% | 31.23% | 41.18% |
| | | Anti | 52.42% | 45.47% | 44.77% | 45.71% | 28.70% | 43.41% |
| | | Proposed | 52.42% | **49.23%** | **49.20%** | **45.86%** | **34.00%** | **46.45%** |
| | WideResNet-28-10 [19] | None | **59.21%** | 33.80% | 33.74% | 31.03% | 40.80% | 39.72% |
| | | Dent | 55.47% | 46.60% | 46.49% | 50.19% | 40.20% | 47.79% |
| | | Anti | 57.60% | 50.46% | 49.98% | 51.11% | 38.80% | 49.59% |
| | | Proposed | 57.78% | **51.78%** | **51.64%** | **51.31%** | **41.60%** | **50.82%** |

**Table 2: AutoAttack on the CIFAR-10/100. The best results are boldfaced, and the second best results are** underlined.

| Dataset | Method | Defense | APGD-CE | APGD-t | FAB-t | Square | AVG |
|---|---|---|---|---|---|---|---|
| CIFAR-10 | ResNet-18 [40] | None | 58.70% | 55.90% | 56.00% | 68.30% | 59.73% |
| | | Dent | 70.25% | 65.50% | 79.40% | 70.85% | 71.50% |
| | | Anti | 78.20% | 77.10% | 83.45% | 67.75% | 76.63% |
| | | Proposed | **79.25%** | **80.10%** | **84.50%** | **76.80%** | **80.16%** |
| CIFAR-100 | PreActResNe-18 [39] | None | 21.00% | 19.35% | 19.70% | 28.75% | 22.20% |
| | | Dent | 36.20% | 30.95% | 44.40% | 36.75% | 37.08% |
| | | Anti | 36.25% | 34.90% | 46.25% | 28.40% | 36.45% |
| | | Proposed | **39.20%** | **40.70%** | **47.95%** | **41.30%** | **42.29%** |

**Table 3: Accuracy of different methods under AA attack and time taken to compute each Batch.**

| Method | WRN-34-10 | Anti | Dent | Proposed w/o SF | Proposed |
|---|---|---|---|---|---|
| Acc.(%) | 62.83% | 67.10% | 64.50% | 71.42% | 72.95% |
| Time(s) | 0.103 | 0.329 | 1.402 | 0.989 | 0.571 |

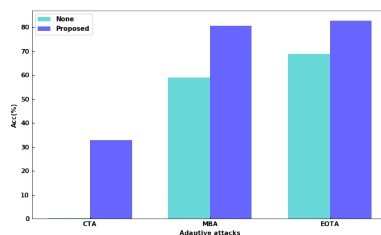

**Figure 5: Results of the proposed method on adaptive attacks.**

## 4.5 Evaluation on AutoAttack

We conducted experiments on the CIFAR-10 and CIFAR-100 datasets using the methods of Sehwag Vikash et al. (ResNet-18) [40] and Rice Leslie et al. (PreActResNet-18) [39], respectively. The experimental results are shown in Table 2, where we can see that our method significantly improves the model's robustness on AutoAttack and outperforms other dynamic defense methods overall. Specifically, on the CIFAR-10 dataset under the APGD-t attack, our method is 9% more robust than Dent, with an average robustness that is 8.66% higher. On the CIFAR-100 dataset, while Anti is more robust than Dent on APGD-CE, APGD-t, and FAB-t, it is ineffective against Square attack, even reducing the original model's robustness (from 28.75% to 28.40%). In contrast, our method not only outperforms Anti on the first three attacks, but also effectively defends against Square attack.

## 4.6 Adaptive attacks on the proposed method

**Continuous transformation attack (CTA).** Dynamic defense is a technique that adjusts itself dynamically based on the input of the current batch. If an attacker continuously attacks the model within the same batch, it is possible that the model will be deflected. Continuous transformation attack is when an attacker continuously attacks the same batch of samples using different methods to shift the model as much as possible. We performed a total of 500 iterations using different attacks: PGD-100→CW-30→PGD-50→CW-50→PGD-20→PGD-50→CW-50→PGD-50→PGD-100.

**Mixed batch attack (MBA).** Optimizing dynamic defense involves applying the same transformation to samples within the

same batch. However, this strategy may be undermined if the batch contains perturbations from different types of attacks. A mixed batch attack uses more than two types of attacks simultaneously, creating multiple types of adversarial samples within the same batch. To address this, we split the samples within a batch into three groups: 30% use PGD-20, 30% use C&W, and the remaining 40% are not attacked.

**Expectation over transformation attack (EOTA).** While the main goal of our method is not to improve robustness by causing gradient confusion in the model, it does result in gradient updates for both the input and model optimization. This renders it impossible for an attacker to obtain deterministic gradients, thus causing the attack to fail. To confirm the robustness of our method, we conducted experiments using EoT [3] in combination with PGD-20.

The experimental results are shown in Figure 5. From the experiments, we can see that our proposed method is robust to various types of adaptive attacks and can guarantee the robustness of the model under different adaptive attacks.

## 4.7 Ablations

**Impact of DIO, DMO and SF on results.** Our investigation focuses on the individual contributions of DIO, DMO and SF. We

**Table 4: Accuracy of different TTA methods on different attacks. The best results are boldfaced, and the second best results are underlined.**

| Method | PGD-20 | C&W | APGD-CE | APGD-t | FAB-t | Square | AVG |
|---|---|---|---|---|---|---|---|
| None | 58.99% | 57.39% | 58.70% | 55.90% | 56.00% | 68.30% | 59.21% |
| Tent | 67.88% | 52.87% | 52.11% | 38.32% | 10.74% | 36.42% | 43.06% |
| Cotta | 62.50% | 57.81% | 62.40% | 60.60% | 77.30% | **81.60**% | 67.04% |
| SAR | 72.69% | 52.68% | 59.00% | 65.30% | 79.00% | 79.15% | 67.97% |
| T3A | 62.27% | 57.53% | 62.00% | 56.20% | 56.40% | 68.80% | 60.53% |
| Proposed | **83.10**% | **83.03**% | **79.25**% | **80.10**% | **84.50**% | 76.80% | **81.13**% |

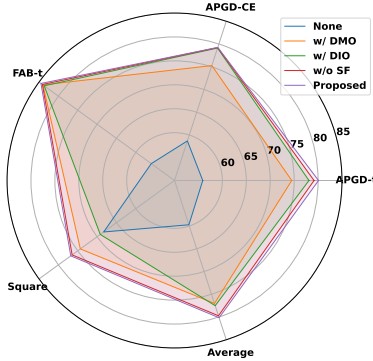

**Figure 6: Results on dynamic input optimization and dynamic model optimization for ablation experiments.**

conducted experiments on the CIFAR-10 dataset using adversarial trained ResNet-18. The results of the experiments are shown in Figure 6, showing five scenarios: without any defense (none), DIO only (w/ DIO), DMO only (w/ DMO), without SF (w/o SF) and the proposed method. Experimental results show that DIO is particularly effective against white-box attacks such as APGD, while DMO is more successful against black-box attacks such as Square. Combining these two approaches can produce better results in terms of enhancing the robustness of the model against various types of attacks. Also, SF reduces the accumulation of errors and improves the robustness of the model.

## 4.8 Time consumption analysis

This subsection presents the time consumption of various methods. For the comparison methods we use ResNet-34-10, Dent (with ResNet-18), and Anti (with ResNet-18). Our method utilizes ResNet-18. The experiments were performed on CIFAR-10. Table 3 shows the accuracy of various methods under AA attack and the time required for inference per batch. Each batch contains 512 samples. The experimental results indicate that while our proposed method incurs additional time consumption, the processing time for multiple samples remains within acceptable limits and does not impose significant delays. For example, our method is not only less time-consuming than Dent, but it also achieves a 8.54% higher accuracy compared to the Dent method. This is because Dent requires iterative optimization six times, whereas our method achieves good results with fewer optimizations. In terms of time consumption, our method incurs an additional 0.242s compared to Anti, but it achieves a 5.85% higher accuracy under AA attack. Importantly,

Anti exhibits instability. It even reduces the robustness of the original model when facing black-box attacks (Square and RayS). Compared to the adversary-trained WideResNet-34-10, our method is in the same order of magnitude as it in terms of time consumption, but its accuracy is 10.12% higher.

## 4.9 Comparison with Test Time Adaptation (TTA) methods.

Our approach is very similar to that of TTA in that both adaptively adjust the model parameters during the inference phase. Therefore, we verify whether TTA's approach is effective in resisting adversarial examples. In our experiments, we applied Tent [47], Cotta [50], SAR [36] and T3A [22] to Sehwag Vikash et al. (ResNet-18) [40]'s proposed methods. We evaluated the effectiveness of these methods against PGD-20, C&W, and AutoAttack as attack methods. Table 4 displays the experimental results. The experimental results indicate that the majority of TTA methods can enhance a model's resistance to adversarial examples to some degree. However, such improvement is limited and may even decrease the model's robustness against certain attacks. While TTA methods account for the reduction in model accuracy caused by distribution bias, adversarial attacks can generate malicious bias that surpasses the adaptive capacity of TTA methods. This can lead to TTA methods failing to defend against adversarial attacks.

## 5 CONCLUSION

In this paper, we introduce a dynamic defense strategy that improves model robustness by optimizing samples through pseudo-label generation. Furthermore, it guides the optimization of the student model using the average augmented predictions from the teacher model. Additionally, we propose a hierarchical optimization approach in model optimization to enhance stability and robustness. Extensive experiments illustrate the effectiveness and applicability of our approach in enhancing model robustness across diverse AT methods, network architectures, and datasets. Although our proposed method substantially enhances model robustness, the computational cost during the inference process escalates due to multiple iterations between attack and defense. In future research, we intend to tackle this issue by investigating cost-effective methods that preserve or enhance robustness.

## ACKNOWLEDGMENTS

The authors would like to thank the anonymous reviewers for their comments. This work was supported by the National Key R&D Program of China under Grant 2023YFB2703800 and the Beijing Natural Science Foundation under Funding No. IS23055. The contact author is Zhen Xiao.

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
