# OpenReview forum: "Embracing Adaptation: An Effective Dynamic Defense Strategy Against Adversarial Examples"
_acmmm.org/ACMMM/2024/Conference — MM2024 Poster_

### Official Review · Reviewer_k154 · 2024-05-16

**Rating:** 5
**Confidence:** 3

**Summary:**

The paper presents a novel dynamic defense approach against adversarial examples in the context of deep learning models. The authors argue that static defense methods are insufficient against adaptive attacks and propose a method that adapts to various attack methods in real-time. The paper utilizes Gaussian Mixture Models (GMM) to obtain structural information of the model, which is combined with model prediction information to generate pseudo-labels for optimizing inputs. Subsequently, it employs information maximization and enhanced mean predictions as optimization objectives, utilizing a hierarchical optimization approach to refine the model. The method is designed to be training-data-agnostic and model-agnostic, making it applicable to a wide range of pre-trained models without the need for retraining.

**Strengths:**

1. Novelty: The paper introduces a dynamic defense strategy that is a significant departure from static defense methods, addressing a critical need in the field of adversarial machine learning.
2. Theoretical Approach: The use of GMM to obtain structural information about the data and optimize the inputs, combined with a hierarchical optimization approach to improve the model, and a more detailed theoretical explanation of the approach, is a theoretically sound and innovative technique.
3. Technical Correctness: The paper provides a detailed explanation of the proposed method, including the mathematical formulation and pseudo-code, which indicates a high level of technical correctness.
4. Adequate Evaluation: The authors conducted extensive experiments comparing their method with state-of-the-art defense methods and various adversarial attacks, as well as ablation experiments, which were critical in determining the effectiveness of the method.
5. Applications: The proposed method's agnosticism to training data and model architecture suggests broad applicability across different models.

**Limitations:**

The dynamic defense strategy proposed in the paper adjusts the input itself and the model parameters according to the current inputs of the model in the inference phase. Does the practice of adjusting model parameters during the inference phase introduce a new attack surface? The authors could briefly discuss this.

**Suitability:**

2

---

### Official Review · Reviewer_cSSd · 2024-05-24

**Rating:** 4
**Confidence:** 3

**Summary:**

This paper presents a novel adversarial attack method. Unlike previous works that only operate on samples, the authors shift the focus to the model itself and propose a dynamic adversarial attack method. Experimental results demonstrate that fine-tuning the model parameters during the attack process is effective.

**Strengths:**

- The idea of updating model parameters during the adversarial attack is innovative.
- The paper provides a detailed analysis of the motivation and methodology, along with comprehensive experimental results that prove the feasibility of the proposed approach.

**Limitations:**

- It would be better to include experiments on other types of datasets. Although conducting experiments on ImageNet is challenging, results from datasets like Tiny-ImageNet could be supplemented.
- The authors could provide a comparison of the attack time and memory usage to comprehensively analyze the method's strengths and weaknesses.
- Would the proposed method still be applicable in a black-box scenario?

**Suitability:**

3

---

### Official Review · Reviewer_X9nn · 2024-05-24

**Rating:** 3
**Confidence:** 2

**Summary:**

The paper "Embracing Adaptation: An Effective Dynamic Defense Strategy Against Adversarial Examples" presents a novel dynamic defense approach to counter adversarial attacks on machine learning models. The proposed method utilizes Gaussian Mixture Models (GMM) for structural data information and pseudo-labels to optimize inputs and model parameters during inference. This training-data-agnostic and model-agnostic strategy significantly enhances model resilience against various adversarial attacks.

**Strengths:**

- Dynamic Adaptation: The method dynamically adjusts inputs and model parameters during inference, improving resilience against evolving adversarial strategies.
- Training-Data and Model Agnostic: It can be applied to pre-trained models without requiring access to training data or retraining.
- Comprehensive Evaluation: Extensive experiments demonstrate the method's superior performance across different types of attacks and datasets.
- Efficiency Improvements: The sample-efficient optimization strategy reduces the number of samples needed for reverse updates, enhancing computational efficiency.

**Limitations:**

I don't quite agree with some of the assumptions about dynamic attacks, and I'd like to know some results. For Mixed batch attacks, will different attacks appear in the same batch and conflict with each other, resulting in reduced attack effectiveness? For example, will a mixed attack (e.g. A 20% + B30%) be worse than a single attack (A50% or B50%) on an undefended model? Regarding the Continuous transformation attack, why should we assume such a cycle? If the type of attack is different from what was predicted, will it be impossible to defend against?

**Suitability:**

2

---

### Meta-Review · Area_Chair_G6ZM · 2024-06-30

**Recommendation:** Accept (Poster)
**Confidence:** 3

**Metareview:**

The paper proposes a novel dynamic defense approach to counter adversarial attacks on machine learning models. After the rebuttal period, it receives mixed final ratings. Two of three reviewers tend to accept this paper, while Reviewer X9nn believes this paper has flaws and is not ready for publication in its current state.

After carefully checking the paper, reviews, and rebuttal document, I decide to accept this paper. Reviewer X9nn also pointed out that the paper could be significantly improved with revisions. The authors are required to revise this paper to address Reviewer X9nn's concerns in the final version.